# *Smilax china* L. Polysaccharide Alleviates Dextran Sulphate Sodium-Induced Colitis and Modulates the Gut Microbiota in Mice

**DOI:** 10.3390/foods12081632

**Published:** 2023-04-13

**Authors:** Xin Li, Gaoxiang Qiao, Lulu Chu, Lezhen Lin, Guodong Zheng

**Affiliations:** Jiangxi Key Laboratory of Natural Product and Functional Food, College of Food Science and Engineering, Jiangxi Agricultural University, Nanchang 330045, China

**Keywords:** *Smilax china* L. polysaccharides, oxidative stress, intestinal inflammation, ulcerative colitis, gut microbiota

## Abstract

This work aimed to investigate the preventive effect of *Smilax china* L. polysaccharide (SCP) on dextran sulfate sodium (DSS)-induced ulcerative colitis (UC) in mice. *Smilax china* L. polysaccharide was isolated by hot water extraction, ethanol precipitation, deproteinization, and purification using DEAE-cellulose column chromatography to yield three polysaccharides: SCP_C, SCP_A, and SCP_N. Acute colitis was induced by administering 3% (*w*/*v*) DSS in drinking water for 7 days. Sulfasalazine, SCP_C, SCP_A, and SCP_N were administered by gavage for 9 days. SCP_C, SCP_A, and SCP_N could significantly improve symptoms, as evidenced by the declining disease activity index (DAI), decreased spleen weight, increased length of the colon, and improved colonic histology. Moreover, SCP_C, SCP_A, and SCP_N increased serum glutathione and decreased the levels of pro-inflammatory cytokines, malondialdehyde, nitric oxide, and myeloperoxidase in colon tissues. Additionally, SCP_C, SCP_A, and SCP_N modulated gut microbiota via ascending the growth of *Lachnospiraceae*, *Muribaculaceae*, *Blautia*, and *Mucispirillum* and descending the abundance of *Akkermansiaceae*, *Deferribacteraceae*, and *Oscillibacter* in mice with UC. The results suggested that *Smilax china* L. polysaccharide ameliorates oxidative stress, balances inflammatory cytokines, and modulates gut microbiota, providing an effective therapeutic strategy for UC in mice.

## 1. Introduction

Inflammatory bowel diseases (IBDs), which include Crohn’s disease (CD) and ulcerative colitis (UC), are relapsing and chronic intestinal inflammations that lead to mucous ulceration and tissue damage and increase the risk of colon carcinogenesis [1]. As a chronic and non-specific IBD, UC is accompanied by weight loss, bloody diarrhea, and abdominal distress [2]. Although the pathophysiology of UC remains unclear, the pathogenesis is multifaceted and includes oxidative stress, genetic predisposition, intestinal barrier dysfunction, and colonic mucosal damage [2,3]. Currently, the treatment of UC generally relies on conventional drugs and biological preparations. For example, sulfasalazine (SSZ) is used to treat IBD and has a broad spectrum of bacteriostatic properties [4]. SSZ can lessen immunological pathogenic damage and inflammation of the intestinal mucosa [5]. Sun et al. [6] used SSZ as a positive control and reported that it alleviated the IBD symptoms. Long-term use of the medications, however, has toxic side effects such as abdominal pain, kidney damage, and hepatotoxicity [7]. Therefore, it is necessary to discover new and safer natural substances to prevent UC. Natural polysaccharides and polyphenols, which are bioactive compounds, are emerging as promising candidates for preventing UC [8].

Gut microbiota, which are referred to as “super-organisms” or the “second genome” of humans, are associated with the onset and progression of UC [9]. Patients with UC have increased intestinal permeability, which causes gut bacteria translocation across the intestinal barrier, exacerbating the occurrence of inflammation and tissue damage [10]. The dysbiosis in UC has been found to involve an alteration in the composition and abundance of bacterial communities, such as a decline in the abundance of *Bacteroidetes* and *Firmicutes* and an increase in *Proteobacteria* [11]. Meanwhile, many studies have indicated a decline in bacterial diversity in UC patients compared to healthy controls [12]. As a result, the regulation of gut microbiota homeostasis has become a new strategy for preventing and palliating UC.

*Smilax china* L., commonly known as “Baqia” and “Jingangteng” (JGT), is mainly derived from the Liliaceae plants that are widely distributed in China. The plant, listed in the Chinese Pharmacopoeia, is a commonly used traditional Chinese herb for the treatment of inflammatory diseases, particularly pelvic inflammatory disease [13]. Several classes of bioactive components, such as stilbenes, flavonoids, polyphenols, polysaccharides, and steroidal saponins, have been isolated from *Smilax china* L. [14]. Previous studies investigating *Smilax china* L. revealed its anti-inflammatory [15], anti-obesity [16], and anti-hyperuricemia [17] activities, as well as other properties. Recently, it has been reported that *Smilax china* L. polysaccharide has anti-inflammatory activity in vitro [18]. However, the protective effects of *Smilax china* L. polysaccharide on UC, particularly in acute colitis, and the role of gut microbiota are poorly understood.

In this work, crude polysaccharide (SCP_C) was isolated and further purified to obtain acidic polysaccharide (SCP_A) and neutral polysaccharide (SCP_N). Then, the chemical characterization and the anti-inflammatory activity of polysaccharide were assessed in an acute model of DSS colitis.

## 2. Materials and Methods

### 2.1. Materials and Reagents

*Smilax china* L. was purchased from Simcere Drugstore in Nanjing, China. The voucher specimen number is B/20190106. DSS (36–50 kDa, reagent grade) was provided by MP Biomedicals (Shanghai China). Sulfasalazine (SSZ) was supplied by Shanghai Sanwei Pharmaceutical Co., Ltd. (Shanghai, China). The DEAE cellulose was purchased from Yuanye Bioengineering Institute (Shanghai, China). Fucose (Fuc), galactose (Gal), rhamnose (Rha), galacturonic acid (Gala), glucose (Glu), xylose (Xyl), and glucuronic acid (Glca) were provided by Sigma Chemical Co. (St. Louis, MO, USA). The AIN-93M diet was purchased from Hunan Silaike Laboratory Animal Co., Ltd. (Changsha, China). Malondialdehyde (MDA), superoxide dismutase (SOD), glutathione (GSH), nitric oxide (NO) and myeloperoxidase (MPO) kits were acquired from Nanjing Jiancheng Bioengineering Institute (Nanjing, China). Enzyme-linked immunosorbent assays (ELISA) for interleukin-6 (IL-6), interleukin-1 beta (IL-1β), interleukin-10 (IL-10), and tumor necrosis factor-α (TNF-α) were purchased from Boster Biological Technology, Co., Ltd. (Wuhan, China). All other chemicals and reagents used in the experiments were of analytical reagent grade.

### 2.2. Isolation and Purification of Polysaccharides

*Smilax china* L. powder was soaked for 24 h in a clean plastic bucket with 2000 mL of ethanol (80%) to remove alcohol-soluble substances, such as small-molecule carbohydrates, lipids, polyphenols, and pigments. After that, the ethanol-soaked residue was air-dried and extracted twice for 2 h with a ratio of 1:20 (*w*/*v*) in 80 °C distilled water. After mixing the extracts, they were then centrifuged at 8400× *g* for 10 min to separate the desired supernatant from the residue. The concentrated liquid was then evaporated in a rotary evaporator until the volume reached 1/3 of the original solution (55 °C). The precipitated polysaccharides were dissolved in hot water, centrifuged, and deproteinated by the Sevag method [19], and then starch was removed with high-temperature α-amylase. Next, combine ethanol at a concentration of 80% (*v*/*v*), and keep the mixture at 25 °C overnight. Centrifugation at 8400× *g* for 10 min was used to collect the precipitate, which was then redissolved in ultrapure water and suspended to remove alcohol at 55 °C. The solution was then dialyzed for 48 h in distilled water with a dialysis bag, followed by 12 h in ultrapure water. Finally, the concentrate was lyophilized in a vacuum freeze dryer to obtain crude polysaccharide (named SCP_C).

SCP_C was further purified using a DEAE-cellulose column (2.6 × 35 cm). Experimental conditions were as follows: SCP_C (6 g, 500 mL) was prepared and poured into a DEAE-resin column for adsorption over 5 h. Sequentially, fractions were eluted with distilled water, 0.05 mol/L NaCl, and 0.1 mol/L NaCl solutions at a flow rate of 1.5 mL/min. The elute was collected and dialyzed in distilled water for 24 h and freeze-dried. Two polysaccharide fractions, named SCP_A (0.05M NaCl eluted) and SCP_N (aqueous eluted), were separated. The fractions, SCP_A and SCP_N, were dialyzed and lyophilized, respectively. The yield of 0.1 mol/L NaCl solution was low, so SCP_A and SCP_N were used in the subsequent studies.

### 2.3. Molecular Weight and Monosaccharide Composition Determination

The molecular weight of SCP_A and SCP_N was determined by high-performance gel permeation chromatography (HPGPC), which was performed in an Agilent 1260 HPLC system equipped with an KS-802 column (8 mm × 300 mm) detected by a refractive index detector. The monosaccharide composition was determined by HPAEC pulsed amperometric detection (PAD), a CarboPac^TM^ PA10 column (2.0 mm × 250 mm), and a Dionex ICS-2500 system. Standard monosaccharides (Fuc, Rha, Ara, Gal, Glc, Xyl, Man, Fru, Gala, and Glca) were used as references.

### 2.4. UV and FT-IR Spectra Analysis

The UV-visible spectra of polysaccharide solution (1 mg/mL) were recorded using an ultraviolet–visible spectrophotometer (model UV-5200PC) in the range of 200–400 cm^−1^. An amount of 1 mg of sample was mixed with 100 mg KBr powder, ground, and then pressed into flakes to record the FT-IR spectra (Nicolet iS5 FT-IR, Thermo Electron Co., Madison, WI, USA) absorbance from 4000–400 cm^−1^ in 64 scans with a resolution of 4 cm^−1^ [20].

### 2.5. Experimental Design

Male BALb/c mice, weighing 20 ± 2 g, were purchased from Hunan Silaike Laboratory Animal Co., Ltd. (Changsha, China SCXK <Xiang> 2016-0002). Mice were free to receive water and food, and they were kept under a standard environment (temperature 24 ± 2 °C, humidity 50 ± 10%, and 12 h light/dark cycle). All experimental involving animals were approved by the Animal Care and Use Committee of Jiangxi Agricultural University (Reg. No. 2018-001). After 5 days of adaptive feeding, all mice were randomly divided into six groups (n = 10): normal chow diet (NCD, received AIN-93M diet), DSS, SSZ (50 mg/kg bw/day), 200 mg/kg bw/day of SCP_C, SCP_A, and SCP_N (the dose was selected based on the results of the pre-experiment). The NCD group was given the same amount of distilled water, and the other groups were given 3% DSS dissolved in distilled water by gavage for 7 days to induce acute colitis. An experimental flow diagram is shown in Figure 2A. Throughout the experimental period, the mice were monitored daily for mental status, water consumption, body weight, fecal consistency, and fecal properties. After that, blood was collected from the orbit under isoflurane anesthesia at the end of the experiment, and the mice were then sacrificed by cervical dislocation. Subsequently, the spleen and colon tissue were quickly excised and weighed from each mouse. The colon length was measured. The caecum contents were collected and then partitioned for future research before storage at −80 °C. The compositions of diet are shown in Appendix A.

### 2.6. Disease Activity Index (DAI) and Colonic Damage

The body weight, stool consistency, and gross bleeding at the anus or in the stool were monitored daily, based on the DAI scoring system in Table 1 [21] and calculating via the formula as follows:DAI = (weight loss score + stool consistency score + gross bleeding score)/3

### 2.7. Histopathologic Analysis of Colon Tissue

For histological analysis, colon tissues were fixed in 4% paraformaldehyde for 24 h. Afterwards, all fixed samples were paraffin embedded, sliced, and stained with hematoxylin eosin (H & E). The colonic structure was observed at 100× magnification by optical microscopy. Histological evaluation was performed according to the reported method [22].

### 2.8. Analysis of the Oxidative Stress Index

The serum superoxide dismutase (SOD) and glutathione (GSH) levels and the contents of malondialdehyde (MDA), nitric oxide (NO), and myeloperoxidase (MPO) in colon tissues were measured using a corresponding assay kit according to the manufacturer’s instructions.

### 2.9. Measurements of Inflammation Cytokines Levels

The colon tissue was weighed and homogenized in a homogenizer in a saline solution. The homogenates were centrifuged 3000× *g* for 15 min at 4 °C. The supernatant was separated, and inflammatory cytokines (including IL-6, IL-10, IL-1β, and TNF-α) were detected using ELISA kits (Boster Biological Technology, Co., Ltd., Wuhan, China).

### 2.10. 16S rDNA Gene Sequencing

Caecum contents were sent to Majorbio Bio-Pharm Technology Co., Ltd., (Shanghai, China) for DNA extraction, PCR amplification, and Illumina MiSeq sequencing. Briefly, microbial DNA was extracted using the DNA extraction kit (D4015, Omega, Inc., Norcross, GA, USA). After amplification and sequencing of the V3–V4 regions of the bacterial 16S rDNA were amplified using primers 341F (5′-CCTACGGGNGGCWGCAG-3′) and 805R (5′-GACTACHVGGGGTATCTAATCC-3′) by a thermocycler PCR system. The program for PCR was 1 cycle of 98 °C for 30 s, followed by 32 cycles of denaturing at 98 °C for 10 s, annealing at 54 °C for 30 s, extension at 72 °C for 45 s, and a final extension at 72 °C for 10 min. Prior to 16S rDNA data analysis, the raw reads were demultiplexed, quality-filtered by fqtrim (V0.94), and merged by FLASH version 1.2.7.

Operational taxonomic units (OTUs) with 97% sequence similarity using Vsearch (v2.3.4), and representative sequences for each OTU were determined through the Ribosomal Database Project classifier. Rarefaction analysis, alpha diversities (ACE, Chao, Shannon, and Simpson indices), and the Weighted UniFrac nonmetric multidimensional scaling (NMDS) Spearman coefficient were performed with the online platform at https://cloud.majorbio.com/ (accessed on 1 June 2021).

### 2.11. Statistics

The data were presented as mean ± standard error of the mean (SEM). Comparisons among different groups were evaluated with one-way analysis of variance (ANOVA) followed by the Dunn’s multiple comparison test and post hoc LSD (SPSS 22, IBM, USA). Different letters (a–d) mean statistical significance (*p* < 0.05).

## 3. Results and Discussion

### 3.1. Structural Analysis of SCP_N and SCP_A

#### 3.1.1. Purification and Identification

Crude polysaccharide was isolated through hot water extraction and ethanol precipitation, deproteinization, dialysis, and drying. The SCP_C extraction yield was 7.1 ± 0.1%. SCP_C was further purified using distilled water, 0.05 M NaCl solution, and 0.1 M NaCl solution, respectively, on a DEAE cellulose column. After concentration and freeze-drying, SCP_N and SCP_A were obtained (Appendix A).

#### 3.1.2. Chemical Compositions of SCP_A and SCP_N

Table 2 shows the contents of total sugar, protein, and uronic acid in SCP_A and SCP_N. SCP_A, which contained 88.5% of total sugar and 24.8% of uronic acid, was composed of Fuc, Rha, Glu, Gal, Xyl, Gala, and Glca with a molecular ratio of 1.01, 3.45, 1.45, 7.41, 0.12, 1.47, and 1.45, respectively. SCP_N, which contained 86.7% of total sugar and no uronic acid, was mainly composed of Fuc, Rha, Glu, Gal, and Xyl with a molecular ratio of 0.18, 0.37, 42.89, 1.11, and 0.79, respectively. The molecular weights of SCP_A and SCP_N were 14.8 kDa and 31.2 kDa, respectively (Table 2).

#### 3.1.3. UV Visible Spectra and FT-IR Spectra

In Figure 1A,B, no UV/vis absorption peak was observed at 260 nm, indicating the absence of nucleic acid in SCP_A and SCP_N. In the UV spectrum, SCP_A and SCP_N showed weak UV absorption at 280 nm, suggesting that they contained a small amount of protein.

The FT-IR spectra of SCP_A and SCP_N are shown in Figure 1C,D. The two absorption peaks at 3412 cm^−1^ and 3400 cm^−1^ corresponded to the –OH stretching vibration. The peaks at 2917 cm^−1^ and 2930 cm^−1^ were for the C–H stretch. Furthermore, the slight absorption peak at 1724 cm^−1^ belonged to carbonyl groups (C=O), the –CH vibration from methyl groups (CH_3_), and the C–O vibration, indicating the presence of O-acetyl groups (–O–COCH_3_) in SCP_A. The absorption peak at 1028 cm^−1^ was caused by the C–C stretching vibration of SCP_A. The peak at 1020 cm^−1^ suggested the presence of a C–O bond in SCP_N. The α-configuration of sugar units was given as 850 cm^−1^ in SCP_A. The characteristic absorption band at 840 cm^−1^ indicated the presence of β-linkage in the structure of the SCP_N polymer.

### 3.2. Polysaccharide Alleviates the Clinical Symptoms of DSS-Induced Colitis

In the DSS-induced UC model, 1 or 2 days after the start of DSS administration, mice were accompanied by symptoms of weakness, dim hair luster, and severe bloody stools in the anus (Appendix A). The weights of mice in the DSS group were significantly lower than those in the SCP_C, SCP_A, and SCP_N groups on the 2nd day. After the 7th day, the body weight of DSS mice consistently decreased and was lower than that of SCP_C- and SCP_A-treated mice (Figure 2B). Figure 2C showed that from day 1, the DAI scores of the DSS-induced groups increased in comparison with those of the NCD group. The DAI score of the NCD group was stable at 0 during the whole experiment, and the growth status performed well. With the development of colitis, the DAI values showed a slow upward trend, manifesting in body weight loss, diarrhea, and bloody stools. A significant difference between the DSS and treatment groups was seen after four days of treatment. Supplementation with SCP_C, SCP_A, and SCP_N reduced the DAI scores. The shortening of the colon and increase in spleen weight are one of the clinical signs of colitis. As displayed in Figure 2D,E, compared with the DSS group, SCP_C, SCP_A, and SCP_N increased the length of colon and decreased spleen weight. The results indicated that polysaccharide could significantly alleviate the clinical symptoms of DSS-induced UC in mice.

After H & E staining, the sections of the colon were subjected to a pathological investigation about the effect of SCP C, SCP A, and SCP N on the severity of inflammation and colon morphology in UC induced by DSS (Figure 2G). DSS treatment caused crypt loss, blurred goblet cells, and inflammatory cell infiltration, resulting in histopathological scores that increased significantly compared to the NCD group. After the intervention with SCP_C, SCP_A, and SCP_N, the DSS-induced pathological alterations are lessened. In particular, SCP_N reduced inflammatory cells’ infiltration and alleviated goblet cell disappearance. Although SCP_A alleviated symptoms to some extent, there was still local inflammatory cell infiltration and goblet cell loss. Histological score analysis also revealed that SCP_C and SCP_N decreased the histopathologic damage (Figure 2F).

### 3.3. Impact of Polysaccharide on Oxidative Stress in DSS-Induced Mice

Inhibition of oxidative stress and improvement of cellular anti-oxidant capacity are effective ways to treat UC [23]. SCP_C, SCP_A, and SCP_N increased the GSH content compared with the DSS group (Figure 3A). There were no significant (*p* > 0.05) differences in the level of SOD between the DSS group and the treatment group (Figure 3B). MDA and MPO activities were used as oxidative damage markers. Excessive MDA induces the release of NO, which leads to an increase in oxygen free radicals and tissue injury, and ultimately triggers inflammatory responses [24]. MPO is an important biomarker of acute inflammation and neutrophil infiltration, and its level rises when colitis occurs [25]. As depicted in Figure 3C–E, SCP_C, SCP_A, and SCP_N decreased oxidative stress as revealed by a reduction in MDA, MPO, and NO levels compared with that of the DSS group. The data indicated that SCP alleviates colon damage caused by oxidative stress.

### 3.4. Effect of Polysaccharide on Inflammatory Cytokine Levels

Cytokines are closely related to the progression of UC, and studies have shown that disease severity is relevant to the balance between pro/anti-inflammatory cytokines. TNF-α and IL-1β, which are primarily released by macrophages and monocytes upon activation have numerous pro-inflammatory properties. Both cytokines are essential for the expansion of intestinal inflammation. TNF-α, in particular, is an important promoter of tissue damage, increasing the production of IL-6 and IL-1β and inducing mucosal inflammation [26,27]. SCP_C, SCP_A, and SCP_N significantly decreased the levels of TNF-α, IL-6, and IL-1β compared with the DSS group (Figure 3F–H). Furthermore, IL-10, an anti-inflammatory cytokine, ameliorates intestinal damage by inhibiting the immune response to autoantigens or foreign antigens and limiting TNF-α, IL-6, and IL-1β secretion [28]. SCP_C, SCP_A, and SCP_N remarkably increased the level of IL-10 as compared with the DSS group (Figure 3I). Accumulating evidence suggests that polysaccharides containing rhamnose and galactose have distinct anti-inflammatory activities, blocking pro-inflammatory cytokines in cell and mouse models [29]. In DSS-induced UC mice, SCP_A and SCP_N composed of Rha and Gal exhibited obvious anti-inflammatory activities. However, the specific molecular mechanism needs to be further investigated. These data indicated that polysaccharide alleviates DSS-induced intestinal inflammation.

### 3.5. Effect of Polysaccharide on the Diversity of Gut Microbiota

The pan/core curves, rarefaction curves, and Shannon index of OTU level analyses showed that the completeness of the sample and the sequencing depth covered rare new phenotypes and diversity (Appendix A). As can be seen from Appendix A, no noticeable difference in microbial diversity was observed between the NCD group and treatment group. Supplementation with SCP_N increased community diversity (Shannon index), compared to the DSS group. Markedly, the Chao1 and ACE indices in the SCP_C, SCP_A, and SCP_N groups were higher than those of the DSS group (Appendix A). Bacterial β-diversity, as assayed by nonmetric multidimensional scaling (NMDS) of the weighted UniFrac index, was then analyzed to depict the structural variability of gut bacterial communities among different groups. The results showed that the gut microbiota of SCP_N group was significantly different from that of the DSS group (Figure 4A). Increasing evidence has indicated that an undesirable alteration of gut microbial diversity and structure is a crucial factor contributing to the occurrence of colonic inflammation [30]. Studies have shown that microbial diversity and richness was decreased in UC patients compared to healthy individuals [12], which is in line with our results. Taken together, the results indicated that polysaccharide significantly increased the gut microbial diversity in DSS-treated mice, which might be partially responsible for its inhibitory effect against UC.

### 3.6. Effect of Polysaccharide on the Structure of Gut Microbiota

*Lachnospiraceae_NK4A136_group*, *norank_f_Muribaculaceae*, *Parabacteroides*, *norank_f_Lachnospiraceae*, *norank_f_Ruminococcaceae*, *unclassified_f_Lachnospiraceae*, *Blautia*, *Akkermansia*, and *Oscillibacter* were the most dominant genera in each group (Figure 4B). Next, we assessed the differences in microbiota composition and relative abundance among different groups at phyla, family, and genus levels.

As shown in Figure 4C, *Firmicutes*, *Bacteroidetes*, *Verrucomicrobia*, and *Deferribacteres* were major phyla in the colonic microbiota. *Firmicutes* and *Bacteroidetes* were found to play an important role in the development of UC [31]. A decline in the phyla *Firmicutes* and *Bacteroidetes* is a predominant signature for dysbiosis in UC patients [32]. Remarkably, *Firmicutes* play an important role in gut homeostasis via the metabolites production, and their abundance promotes the anti-inflammatory and anti-tumorigenic properties [33]. SCP_N treatment effectively increased the abundance of *Firmicutes* and *Bacteroidetes* (Figure 4D,E). One homogeneous polysaccharide from *Scutellaria baicalensis Georgi*, which is mainly composed of mannose, ribose, rhamnose, glucuronic acid, glucose, xylose, arabinose, and fucose, improved colitis by increasing the growth of *Firmicutes*, *Bifidobacterium*, *Lactobacillus*, and *Roseburia* [34]. Additionally, polysaccharides from soybean residue fermented by *Neurospora crassa*, which mainly contained rhamnose, arabinose, fucose, mannose, glucose, and galactose, increased the growth of *Firmicutes* in DSS-induced colitis [35]. Consistent with previous findings, the current results showed that SCP_N, which primarily contained fucose, rhamnose, glucose, galactose, and xylose, prevented gut dysbiosis in DSS-induced UC mice. As shown in Figure 4F, at the family level, the gut microbiota was mainly composed of *Lachnospiraceae*, *Deferribacteraceae*, *Muribaculaceae*, *Akkermansiaceae*, and *Ruminococcaceae*. In this study, SCP_A and SCP_N significantly decreased the abundance of *Akkermansiaceae* in UC mice (Figure 4G). *Akkermansiaceae*, which belongs to *Verrucomicrobia*, one of the mucin-degrading specialists in the gut, was reported to be increased in mice with DSS-induced colitis and in patients suffering from colorectal cancer [36]. Intestinal mucins are degraded after *A. muciniphila* overgrowth, eventually leading to colitis deterioration [37]. Its potential role in the development of colitis remains controversial. *A. muciniphila* was found to be a potent anti-inflammatory intestinal bacterium that adhered to intestinal enterocytes, strengthened the integrity of the epithelial cell layer, and was inversely correlated with inflammation [38]. Considering its mucin-degrading properties, we speculated that *A. muciniphila* invaded and degraded the colon mucosa, which led to an increased permeability and access to other microbes that cause inflammation. Moreover, SCP_A and SCP_N increased the growth of *Lachnospiraceae* in DSS-induced mice (Figure 4H). It was reported that *Lachnospiraceae* is a probiotic that can induce an accumulation of T reg cells in the colon, which is involved in the generation of butyrate and reduces inflammation [39]. As shown in Figure 4I,J, compared with the DSS group, the relative abundance of *Muribaculaceae* in the SCP_N group significantly increased, and that of *Deferribacteraceae* significantly decreased. SCP_N drastically elevated the growth of the *Muribaculaceae* in DSS-induced mice. Specific bacteria of the *Muribaculaceae*, which belongs to the Bacteroidetes, is related to the intestinal barrier function, which regulates immune cells to decrease the production of pro-inflammatory cytokines [40,41]. *Muribaculaceae* has been detected as the dominant gut microbiota in healthy humans and animals and is linked to the production of acetate and propionate [42].

At the genus level, compared to the DSS group, the relative abundance of *Lachnospiraceae_NK4A136_group*, *Blautia*, *norank_f_Muribaculaceae*, *Erysipelatocostridium*, and *Rikenellaceae_RC9_gut_group* significantly increased in SCP_A and SCP_N, while that of *Eubacterium_fissicatena_group*, *norank_f_Clostridiales_vadinBB60_group*, *norank_f_Ruminococcaceae*, *Oscillibacter*, *Mucispirillum*, and *Lactobacillus* significantly decreased (Figure 5). SCP_A and SCP_N treatment significantly increased the abundance of *norank_f_Muribaculaceae*, and the genera have beneficial effects on the prevention of IBD [43]. In addition, as a kind of key probiotic, the *Lactobacillus* genera may be beneficial during DSS-induced UC [44]. In the present study, the amount of *Lactobacillus* on colonic mucosa was decreased (Figure 5), and similar results have been obtained in other studies [45]. Notably, SCP_C, SCP_A, and SCP_N administration drastically inhibited the growth of *Oscillibacter* in DSS-induced mice. *Oscillibacter* has been recognized as a newly discovered genus associated with digestive diseases and the severity of DSS-induced UC [46]. The genus *Lachnospiraceae_NK4A136_group*, which has anti-inflammatory properties [47], increased significantly after SCP_A and SCP_N treatment, and is negatively correlated with MPO, DAI, and TNF-α. Meanwhile, SCP_A and SCP_N significantly increased the abundance of *Blautia* in UC mice. *Blautia* is one of the major intestinal microbes. The richness of *Blautia* has been linked to colorectal cancer and IBDs [48]. Together, these findings demonstrated that SCP_C, SCP_A, and SCP_N intervention played a key role in the regulation of the gut microbiota in DSS-induced colonic inflammation.

### 3.7. Correlation between UC-Related Parameters and Key Phylotypes of Microbiota

Results of db-RDA showed a significant correlation between microbiota and several pro-inflammatory cytokines, NO, MPO, MPA, and IL-10. The degree of correlations between UC-related biochemical parameters and microbiota was as follows: MDA > TNF-α > IL-1 β > IL-6 > NO > MPO (Figure 6A).

Spearman’s correlation analysis was used to determine the relationship between UC-related parameters and particular microorganisms (Figure 6B). *Marvinbryantia*, *A2*, *Lachnospiraceae_NK4A136_group*, *Sporosarcina*, *Oceanisphaera*, and *Roseburia* were negatively correlated with MPO, DAI, and TNF-α, and they were positively correlated with IL-10 and colon length. *Blautia*, *Saccharofermentans*, *Succiniclasticum*, *Prevotella_1*, and *Ruminococcaceae_NK4A214_group* were negatively correlated with IL-1β. *Eubacterium]_fissicatena_group* and *Akkermansia* were positively correlated with MPO, DAI, and TNF-α. GCA-*900066225*, *Eubacterium]_coprostanoligenes_group*, *Harryflintia*, *Lachnospiraceae_UCG-006*, and Rumini*clostridium_6* were positively correlated with TNF-α.

## 4. Conclusions

In summary, different polysaccharides, namely, SCP_C, SCP_A, and SCP_N from *Smilax china* L., were isolated and purified, and their chemical structures were determined by HPGPC, HPAEC-PAD, UV -Vis, and FTIR. The in vivo anti-inflammatory effect of polysaccharide was investigated, and an acute colitis model was induced by 3% DSS. The basic symptoms observed in this model, including body weight loss, diarrhea, and bloody stools reflect the degree of severity. Supplementation with SCP_C, SCP_A, and SCP_N decreases DAI scores, alleviates colon inflammation and oxidative stress, and modulates the gut microbiota, thereby ameliorating DSS-induced colitis. However, the structure–activity relationship of SCP_C, SCP_A, and SCP_N in the treatment of UC needs further study. Our findings indicate that SCP_C, SCP_A, and SCP_N have strong potential for preventing and treating UC.

## Figures and Tables

**Figure 1 foods-12-01632-f001:**
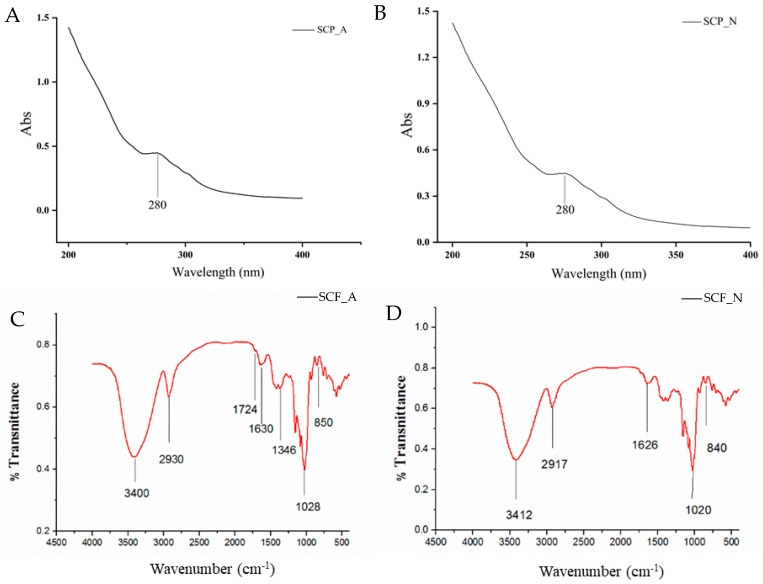
Spectrum of *Smilax china* L. polysaccharide. (**A**) UV spectrum of SCP_A; (**B**) UV spectrum of SCP_N; (**C**) FT-IR spectra of SCF_A; (**D**) FT-IR spectra of SCF_N.

**Figure 2 foods-12-01632-f002:**
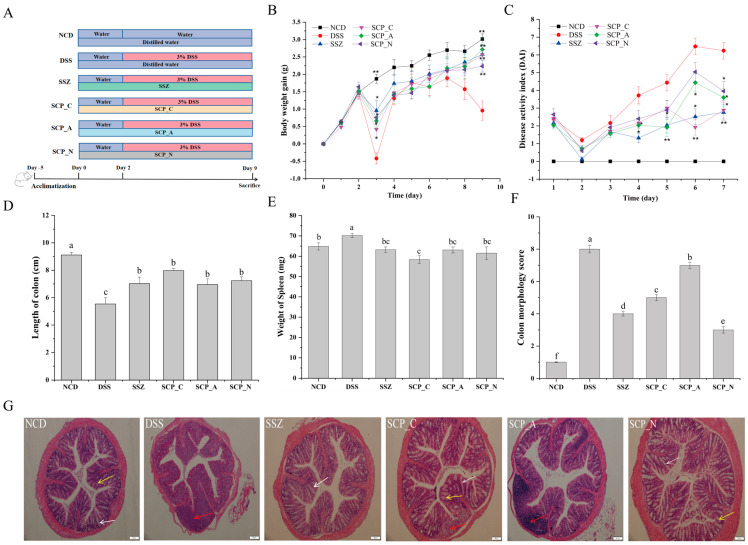
Polysaccharide alleviated symptoms of DSS-induced UC in mice. (**A**) Experimental design; (**B**) Body weight gain; (**C**) DAI scores; * *p* < 0.05, ** *p* < 0.01 vs. DSS. (**D**) Colon length; (**E**) Spleen weight; (**F**) Colon morphology score; (**G**) H & E staining microscopic image of colon tissue. The red, yellow, and white arrows indicate the locations of inflammatory infiltration, goblet cells, and crypts, respectively (scale bars 100 μm). Different letters (a–f) represent significant differences among groups (*p* < 0.05).

**Figure 3 foods-12-01632-f003:**
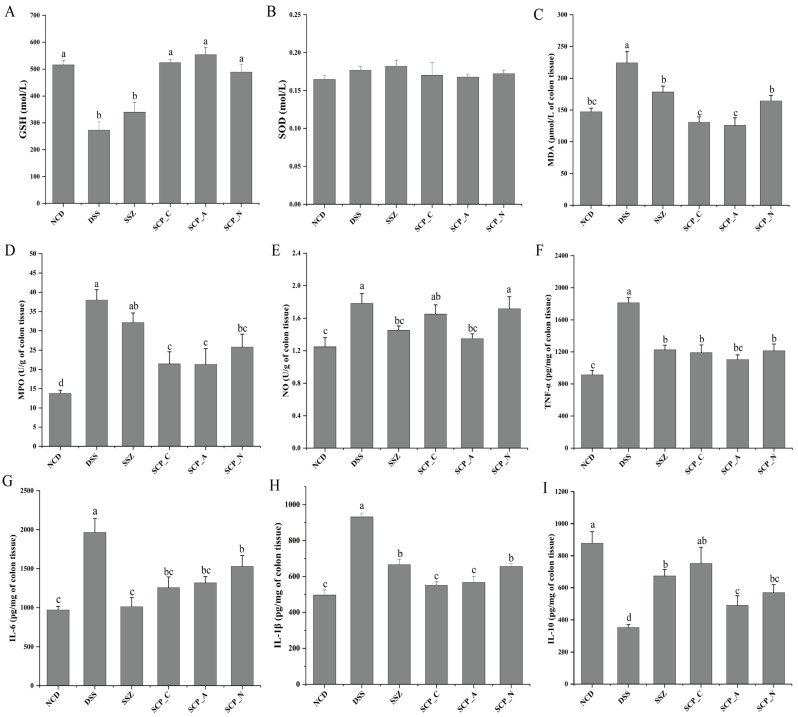
Effect of polysaccharide treatment on oxidative stress and inflammatory cytokines. (**A**) GSH and (**B**) SOD in serum; the levels of (**C**) MDA, (**D**) MPO, (**E**) NO, (**F**) TNF-α, (**G**) IL-6, (**H**) IL-1β, and (**I**) IL-10 in colon tissues. Different letters (a–d) represent significant differences among groups (*p* < 0.05).

**Figure 4 foods-12-01632-f004:**
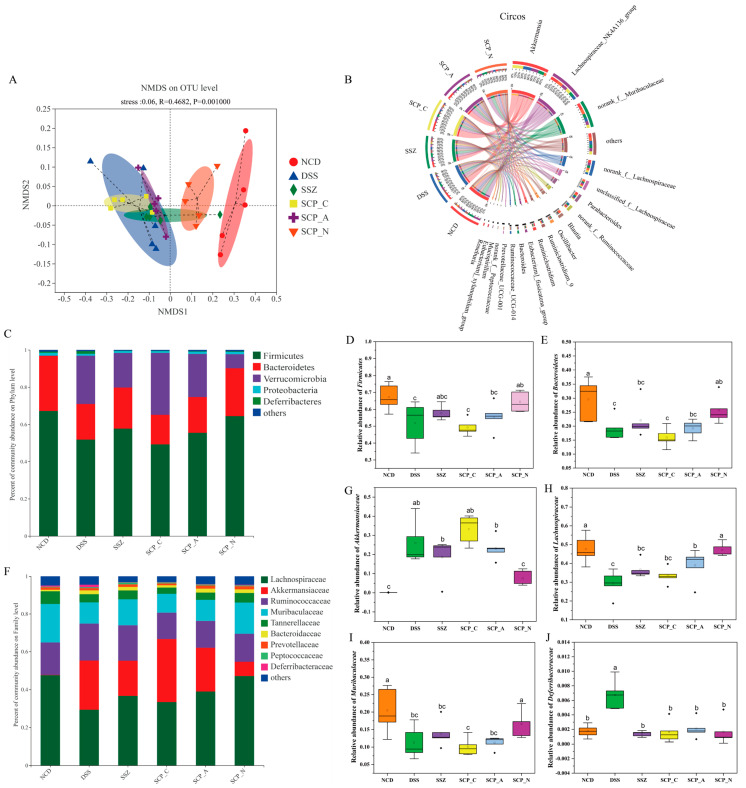
Effects of polysaccharide on gut microbiota composition of different groups. (**A**) NMDS score plot in OTU level; (**B**) Linear relationships between samples and species; (**C**) Bacterial taxonomic composition at the phylum level; (**D**) relative abundance of Firmicutes; (**E**) relative abundance of Bacteroidetes; (**F**) Bacterial taxonomic composition at the family level; relative abundance of (**G**) *Akkermansiaceae*, (**H**) *Lachnospiraceae*, (**I**) *Muribaculaceae*, and (**J**) *Deferribacteraceae*. Different letters (a–c) represent significant differences among groups (*p* < 0.05).

**Figure 5 foods-12-01632-f005:**
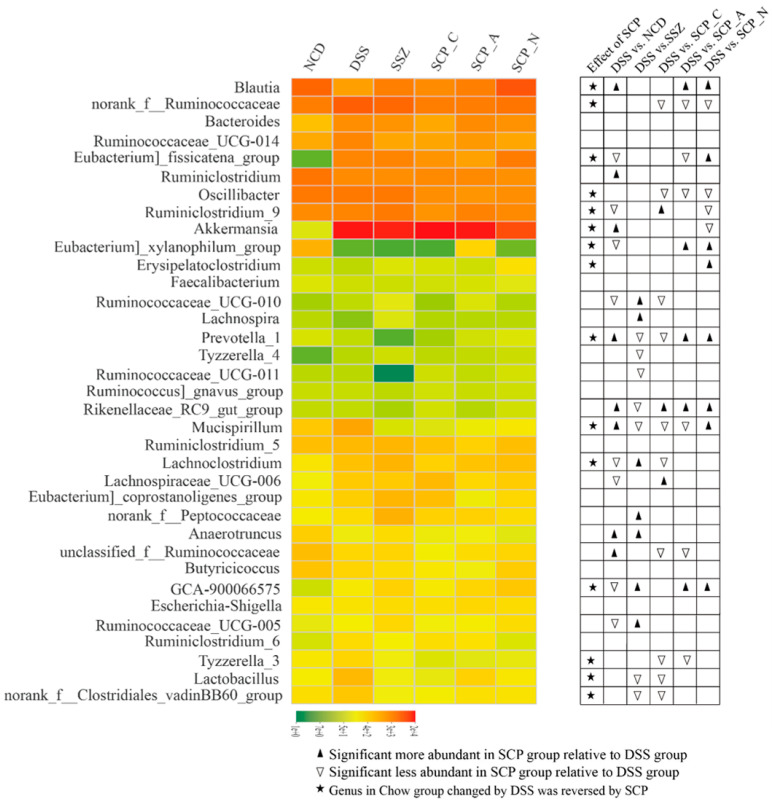
The relative abundance of the top 35 genera in given NCD, DSS, SCP_C, SCP_A and SCP_N was included in the heatmap. Statistical significance was determined using one-way ANOVA with Dunn’s multiple comparison test.

**Figure 6 foods-12-01632-f006:**
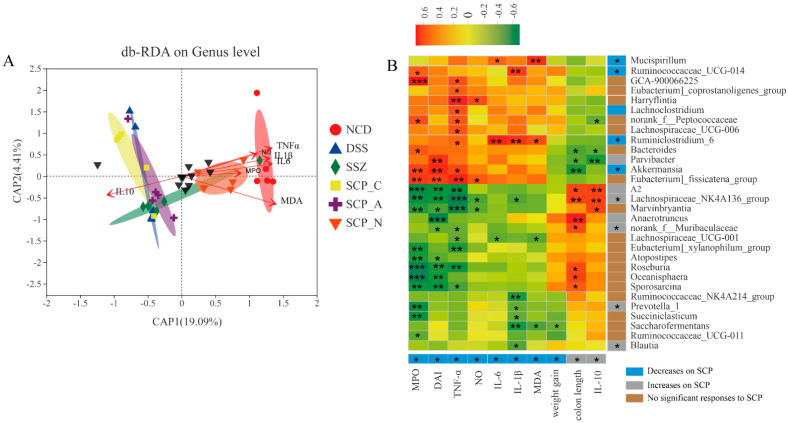
Correlations between UC-related parameters and key phylotypes of microbiota. (**A**) Distance-based Redundancy Analysis (db-RDA) plot showing the relationship between UC-related parameters and gut microbiota structure. The plots represent a db-RDA ordination using the Bray–Curtis distance. The black triangles represent the top ten species. (**B**) Heatmap of the Spearman correlations between the gut microbial composition with UC-related indices. The asterisk (*) denotes statistical significance for the correlations between phenotypes and genus abundance. * 0.01 < *p* ≤ 0.05, ** 0.001 < *p* ≤ 0.01, *** *p* ≤ 0.001.

**Table 1 foods-12-01632-t001:** DAI evaluation criteria.

Score	Weight Loss (%)	Stool Consistency	Occult/Gross Bleeding
0	<1	normal	negative
1	1~5	loose	weakly positive
2	6~10	loose stool	positive
3	11~15	very soft, wet	strong positive
4	>15	watery diarrhea	blood traces in stool visible

**Table 2 foods-12-01632-t002:** Chemical composition of SCP_A and SCP_N.

	Properties						
	Total Sugar/%	Protein/%	Uronic Acid/%	*Mw*/kDa			
SCP_A	88.5 ± 0.6	4.1 ± 1.0	24.8 ± 1.4	14.8			
SCP_N	86.7 ± 0.7	2.8 ± 0.7	n.d. ^a^	31.2			
	**Molar Ratio**						
	**Fuc**	**Rha**	**Glu**	**Gal**	**Xyl**	**Gala**	**Glca**
SCP_A	1.01	3.45	1.45	7.41	0.12	1.47	1.45
SCP_N	0.18	0.37	42.89	1.11	0.79	n.d. ^a^	n.d. ^a^

^a^ n.d., not detected.

## Data Availability

The data presented in this study are available on request from the corresponding author.

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
