# Peer review of "Smilax china L. Polysaccharide Alleviates Dextran Sulphate Sodium-Induced Colitis and Modulates the Gut Microbiota in Mice"

_foods, 2023, doi:10.3390/foods12081632_

Round 1

Reviewer 1 Report

This study showed that Smilax china L. polysaccharide is useful in the treatment of UC by ameliorating oxidative stress, balancing inflammatory cytokines, and modulating the gut microbiota.

Overall, I felt that this was an interesting study.

However, minor modifications would be needed. See below.

The font in the figure is too small and will confuse the reader.

The results in Figure 2, E, should be indicated with color-distinguished arrows, since it is unclear which part of the image is being shown and discussed.

The reason for using sulfasalazine should be briefly stated in abstract.

Line 85-87

Experimental conditions for the flow amount of each solvent should be indicated.

The ledgend of fig.1, method (Line 100-101) and result (Line 170-171) differ in terms of UV spectrum and UV-Vis spectrum. In addition, the units of spectral measurement range will be nm.

Line 142-143

Information on the nucleotide sequences of the 341F and 805R primers and PCR conditions should be provided.

The following should be discussed

Did changes in gut bacterial diversity suppress UC?

Did the suppression of UC cause changes in gut bacterial diversity?

Author Response

Reviewer #1

This study showed that Smilax china L. polysaccharide is useful in the treatment of UC by ameliorating oxidative stress, balancing inflammatory cytokines, and modulating the gut microbiota. Overall, I felt that this was an interesting study. However, minor modifications would be needed. See below.

Response: Thank you very much for your great support for the acceptance our manuscript in foods. According to your suggestions, we have carefully revised the manuscript. Your valuable suggestions will be of great help to us.

General comments

The font in the figure is too small and will confuse the reader.

Response: Thanks for your constructive comments and suggestions. The font in the figures has been improved to make the figure clearer.

The results in Figure 2, E, should be indicated with color-distinguished arrows, since it is unclear which part of the image is being shown and discussed.

Response: Thanks for your suggestion. Figure 2E was indicated with color-distinguished arrows.

The reason for using sulfasalazine should be briefly stated in abstract.

Response: We sincerely appreciate your advice. We have added a description of this section. Please refer to lines 36-39.  

Line 85-87. Experimental conditions for the flow amount of each solvent should be indicated.

Response: The crude polysaccharide was further purified using a DEAE-cellulose column. The experimental conditions were as follows: SCP_C (6g, 500mL) was prepared and poured into a DEAE-resin column for adsorption 5 hours. Sequentially was eluted with distilled water, 0.05 mol/L NaCl and 0.1 mol/L NaCl solution at a flow rate of 1.5ml/min. The collected fractions were dialyzed in distilled water for 24 h and freeze-dried.

The ledgend of fig.1, method (Line 100-101) and result (Line 170-171) differ in terms of UV spectrum and UV-Vis spectrum. In addition, the units of spectral measurement range will be nm.

Response: Thank you for your correction. It was an oversight on our part. The units of UV spectrum and UV-Vis spectrum have been corrected.

Line 142-143: Information on the nucleotide sequences of the 341F and 805R primers and PCR conditions should be provided.

Response: We have carefully revised it according to your suggestions. The description of this section was added. The details are as follows:

341F (5’-CCTACGGGNGGCWGCAG-3’) and 805R (5’-GACTACHVGGGGTATCT

AATCC-3’). The program for PCR was 1 cycle of 98°C for 30s, followed by 32 cycles at 98°C for 10s, annealing at 54°C for 30s, extension at 72°C for 45s and a final extension at 72°C for 10 min.

The following should be discussed

Did changes in gut bacterial diversity suppress UC?

Response: Thanks for your proposal. A discussion of the relationship between changes in gut bacterial diversity and UC has been added to the manuscript.

Did the suppression of UC cause changes in gut bacterial diversity?

Response: We sincerely appreciate your advice. A discussion on whether suppression of UC causes changes in gut bacterial diversity has been added to the manuscript.

Reviewer 2 Report

Dear authors,

please find my comments to your manuscript in attached pdf file. 

Author Response

Reviewer #2

Li et al have investigated whether a bioactive component from a Chinese herb, Smilax china L, could affect DSS induced colitis in mice. They specifically looked at polysaccharide from Smilax china L, as it was previously shown to have anti-inflammatory properties in vitro. They have isolated and characterized several fractions and tested these in an acute model of ulcerative colitis, so test whether these fractions could alleviate the colitis symptoms and to evaluate their effect on the microbiota. Their obtained results show indeed alleviation of the symptoms, lower disease scoring and lower tissue damage. Furthermore, they show a change in microbiota. The presented work is of value, as it is desired to find alternative ways to alleviate symptoms of IBD. However, there are some issues in this work that need to be addressed. In general, there are quite some issues that the authors need to elaborate on in their discussion.

Response: Thank you for taking the time to read and consider our manuscript. We appreciate your comments on this study. We have carefully revised the manuscript according to your comments to make it more suitable for the foods.

General concept comments:

- The authors should be more clear on the diet they have used in this study. In section 2.5 they mention normal chow diet, but in supplemented material they mention AIN93 diet. There is a big difference in these diets, and the outcome and severity of the colitis model used could really differ based on which diet was used. Also, is it not clear whether the other groups received the same diet, either chow or AIN93. This information should be included, and possibly some speculation of this issue on the results in the discussion.

Response: Thanks for your comments. We apologize for the misunderstanding due to our confusion. The NCD group was fed AIN-93M diet, and all the experimental diets were AIN-93M diet. This information has been added to the manuscript.

- 5 days of adaptation is rather short for microbiota adaptation. This is quite an issue especially since this paper focusses on the microbiota in quite some detail. What diet did the mice get before the start of this study? If they were on the same chow/AIN93 diet that was continued in this study except for the additional gavages than it should not   be an issue. The authors need to clarify this, and mention this as a possible flaw in their discussion when indeed the mice were on a different diet before this study started.

Response: Thank you for your comments. Five days of adaptive feeding were used to adapt mice to the experimental environment after changing the feeding environment.  Furthermore, all mice were fed the AIN-93M diet for 14 days. We have added a description of this section to the method section.

- Section 2.5: Did all animals receive gavages? Also the control animals in group NCD? What was the solvent of the gavages? This information should be stated in the manuscript.

Response: Thanks for your comments. All animals were given gavage, including the NCD group. The NCD group was given the same volume of distilled water throughout the experiment. This information has been added to the methods section. 

- Section 2.5: Blood was isolated, what was done with this? How was it processed, was plasma or serum isolated?

Response: Thanks for your comments. Blood was collected from orbit, centrifuged 3500g at 4 °C for 10 min, serum was separated and collected, and stored at –80°C until analysis.

- Crucial information on the 16S sequencing and analysis is missing. This includes for example the QIIME version (note that is this is not >2, this is currently no longer acceptable), ASV picking strategy, taxonomic classification method, reference database, b-div rarefaction analysis. This information is essential. Also in the statistics part crucial information is lacking, like multiple testing correction strategy.

Response: We are really appreciated for your kind suggestion. In response to your suggestion, we provided more information on bioinformatics analysis of sequencing data. The details are as follows:

Caecum contents were sent to Majorbio Bio-Pharm Technology Co. Ltd, (Shanghai, China) for DNA extraction, PCR amplification and Illumina MiSeq sequencing. Briefly, microbial DNA was extracted using DNA extraction kit (D4015, Omega, Inc., USA). After amplification and sequencing of V3–V4 regions of bacterial 16S rDNA were amplified using primers 341F (5’-CCTACGGGNGGCWGCAG-3’) and 805R (5’-GACTACHVGGGGTATCTAATCC-3’) by a thermocycler PCR system. The program for PCR was 1 cycle of 98°C for 30 s, followed by 32 cycles of denaturing at 98°C for 10s, annealing at 54°C for 30s, extension at 72°C for 45s and a final extension at 72°C for 10 min. Prior to 16S rDNA data analysis, the raw reads were demultiplexed, quality-filtered by fqtrim (V0.94) and merged by FLASH version 1.2.7.

Operational taxonomic units (OTUs) with 97 % sequence similarity using Vsearch (v 2.3.4), and representative sequences for each OUT were determined through Ribo-somal Database Project classifier. Rarefaction analysis, the community richness index and alpha diversities, and the Weighted UniFrac non-metric multidimensional scaling (NMDS) Spearman coefficient were performed with the online platform at https://cloud.majorbio.com/.  

- The authors show in line 169 the size of their isolated fractions, but this reviewer misses some elaboration on previously found data. How do their compounds compare to others found to be effective in literature?

Response: Thanks for your proposal. We have already discussed the relevant content in the manuscript. Please refer to lines 324-331.

- In line 193 the authors mention that the body weight of the DSS mice consistently decreased, but this is not clear from the figure they mention. Could they please elaborate on this? In the figure one could only see a continuous decrease in body weight in the DSS group from day 7 onwards

Response: Thanks for your correction. This was our negligence, and the DSS group continued to lose weight after 7 days. These mistakes were corrected in the revised manuscript in red fonts.

- Based on the images shown the authors claim to see goblet cell exhaustion. Based on the images shown here using this staining this cannot be concluded. Furthermore, they say in line 200 that they see ‘alleviated goblet cell reduction’, these claims cannot be made based on these results, and only H&E staining. It would be very advantageous if they could place arrows in the image showing what they mention (crypt abscess, edema etc). Also, this reviewer thinks the authors should include a scoring of the histological slides, using a colitis scoring system (for example the one described in Nature Protocols, Wirtz et al, 2017, vol 12 no 7, table 4). This would strengthen their observations.

Response: Thank you for your advice. According to your opinions, we have carefully read the relevant literature you provided, and added relevant information. The suggestions and references provided greatly improved our manuscripts. Furthermore, Figure 2E was represented by color-distinguished arrows.

- Figure 2c: how do the authors explain a DAI of when the animals did not yet receive any DSS (day 1)? The control animals have a DAI of 0. Does this mean that there is an effect of the gavages? Like body weight loss due to stress? But then: does the NCD group not receive any gavages? The authors need to elaborate on this.

Response: Thank you for the comments. DAI is determined by daily assessing changes in body weight, stool consistency, and gross bleeding at the anus or in the stool. In this study, SSZ, SCP_C, SCP_A, SCP_N were administered by gavage for 9 days. After 2 days of gavage, mice were given 3% (w/v) DSS for 7 days. DSS caused a weight decrease (starting at day 2) and DAI score increase (starting at day 1). In addition, the NCD group was given the same volume of distilled water throughout the experiment, the growth status was good, and DAI score of NCD group was stable at 0 during the experiment. We have added a description of this section in lines 230-240 of the manuscript.

- Have the authors also looked at the spleen length and weight? And at serum cytokines, or only at colon cytokines? This could tell more about the systemic inflammation.

Response: Thanks for your proposal. I’m sorry, we didn’t measure the length of the spleen, the weight of the spleen has been added in Figure 2. Additionally, we only measured colon cytokines. In future studies, we will take your suggestion and also measure serum cytokines. 

- Did the authors look at the effect of SCP on mucus production? (for example by qPCR). Same for antimicrobial peptides? Could the SCP have an effect on either of these and thus indirectly on the microbiota? Could the authors either perform these tests of at least elaborate on this in their discussion? Because based on the data shown in this manuscript the claim in line 305 cannot be made as such.

Response: I’m sorry that we didn’t examine the effect of SCP on mucus production and antimicrobial peptides. Thanks for your proposal. We have revised line 305 to make this statement more reasonable. And the relevant problems were analyzed and discussed.

- This reviewer is not convinced that the authors can distinguish between stopping tissue damage and preventing tissue damage like they claim in their conclusion section. Consider rewording this.

Response: Thanks for your suggestion and correction. In order to increase the readability of the manuscript, English sentences have been modified and rewritten.

- Data availability statement: it is essential to deposit the data (including the raw sequencing reads) in the appropriate repositories. So stating that this is not applicable is not acceptable.

Response: Thanks for your comments. The data availability statement has been corrected. The data presented in this study are available upon request from the corresponding author.

Specific comments:

- Line 20: add: in mice at the end of the sentence

Response: Thanks for your suggestion. It was added.

- Line 42: what are fungal bacteria? I suggest to rewrite this sentence

Response: Thanks for your suggestion. It has been rewritten in the revised manuscript in red fonts. Please refer to lines 48-52.

- Line 53: in vitro should be italics

Response: Thank you for the comments. It has been corrected

- Line 58: add to the end of the sentence: ‘were assessed in an acute model of DSS colitis’ for clarity.

Response: Thanks for your kind suggestion. We have added a description ‘were assessed in an acute model of DSS colitis’.

- Line 63: which molecular weight DSS was used? Is essential to know for the outcome of the experiment

Response: Thanks for your suggestion. We added DSS molecular weight (36–50 kDa, reagent grade) in the Materials and Reagents section.

- Line 64: purposed should be purchased I think?

Response: Thanks for your correction. We have carefully corrected the mistakes and shortcomings in the manuscript.

- Part 2.2 is not completely clear to me. Line 82 contains a grammatical incorrect part, but also in line 86 is it not completely clear, should there be and/or in between the H2O and NaCl? Later, in the result sections (line 158 and 159), it becomes more clear what was done here, but not from this section. Perhaps rewrite this a bit to make it more clear

Response: Thanks for your correction. We have carefully corrected the mistakes and shortcomings in the manuscript. English spelling, grammar and usage were corrected and edited by a native English speaker.

- Line 114 is not clear: was the water supplemented, or did they use distilled water supplemented with DSS? I assume the latter

Response: Thanks for your correction. It has been corrected.

- Line 118: mention which anaesthesia was used

Response: Thanks for your comment. Isoflurane was used to anesthetize mice.

- Line 137: how was the tissue homogenized? Crushing under liquid nitrogen, using an ultraturrax, etc?

Response: Thanks for your comments. Colon tissue was homogenized by a homogenizer. We have been added the manuscript.

- Line 153: results and discussion

Response: Thanks for your correction. It has been corrected.

- Line 190: change ‘after DSS’ to ‘after the start of DSS administration’

Response: Thanks for your correction. It has been corrected.

- Line 192: if something is significant, why is this not shown in the figure?

Response: Thanks for your comment. It is our negligence, and the significant difference has been added to the figure.

- Line 198: which model mice are meant here? This is not clear from the text

Response: Thanks for your comment. The model mice here refer to UC mice, and we used male BALb/c mice.

- Line 201: not sure this claim can be, and should be made, that a stable model was established

Response: Thanks for your correction. It has been corrected

- Figure 2B and C: were there no significant differences here?

Response: Thanks for your comment. Figure 2B and C were marked with * to indicate the significant differences.

- Line 248: UC mice is DSS group? Compared to what was it improved? Keep the same wording everywhere for clarity reasons.

Response: Thanks for your comment. UC mice are the DSS group, we have carefully revised the manuscript based on your suggestions.

- Line 254: how do the authors define dramatically? Consider using a different word here

Response: We sincerely appreciate your advice. We used “significantly” to replace “dramatically”.

- Line 257: it is not really clear what is meant here.

Response: Thanks for your comment. We realized that some of the expression in our manuscript can be confusing. We carefully revised the manuscript, and it was revised by a native English speaker.

- Line 268: Zhang reported on polysaccharide. Which one? The same one as used in this paper? The whole fraction? Please specify

Response: Thanks for your comments. The polysaccharide Zhang reported is different from ours, which may cause your misunderstanding. After reviewing the literature, we considered deleting this literature and adding new relevant literature. We rewrote this sentence.

- Line 274: What do the authors mean here? In the mucinous layer of degradable

Response: Thanks for your comment. We realized that some expressions of our manuscript can be confusing. We carefully revised the manuscript to make our paper clearer.

- Line 296: the amount of Lactobacillus on colonic mucosa were suppressed. This claim can only be made when the authors would have looked at mucosal samples. This was not done, they have looked at fecal samples. Thus this should be worded differently here

Response: Thanks for your correction. We carefully revised the manuscript based on your suggestions.

- Figure 5: was any multiple testing correction applied here?

Response: Thanks for your comment. Statistical significance was determined using a one-way ANOVA with Dunn’s multiple comparison test. * p < 0.05, ** p < 0.01, and *** p < 0.001. It has been added to the Figure 5 caption. 

- Figure 6: The taxa triangles without any labels serve no purpose. In addition: how was ‘structure’ defined (i.e. what distance metric)?

Response: Thank you for your comment. The db-RDA (distance-based redundancy analysis) plot showed the relationship between inflammatory cytokines, oxidative factors, and microbial community structure. The plot represents db-RDA ordination based on Bray-Curtis distance. The black triangle represents the top eight species. Visualization of correlation networks according to partial correlation differences between cecal microbiota, and parameters associated with inflammatory and oxidative factors.

Reviewer 3 Report

The manuscript is adequate and interesting, The format is correct and in general is well written and results expresed as clarity. 

I only suguuest the authors to give more information about the bioinformatic analysis of the sequencing data (quality of sequences, denoising read, rarefactoon depth, etc. And also to include the reference for qiime and the version employed. 

Author Response

Reviewer 3#

The manuscript is adequate and interesting, the format is correct and in general is well written and results expresed as clarity.

Response: Thank you very much for your great support for acceptance our manuscript in foods.

I only suguuest the authors to give more information about the bioinformatic analysis of the sequencing data (quality of sequences, denoising read, rarefactoon depth, etc. And also to include the reference for qiime and the version employed.

Response: We are really appreciated for your kind suggestion. According to your suggestion, we have provided more information about the bioinformatic analysis of the sequencing data (quality of sequences, denoising read, rarefactoon depth, etc. and also to include the reference for Qime and the version employed. The details are as follows:

Caecum contents were sent to Majorbio Bio-Pharm Technology Co. Ltd, (Shanghai, China) for DNA extraction, PCR amplification and Illumina MiSeq sequencing. Briefly, microbial DNA was extracted using DNA extraction kit (D4015, Omega, Inc., USA). After amplification and sequencing of V3–V4 regions of bacterial 16S rDNA were amplified using primers 341F (5’-CCTACGGGNGGCWGCAG-3’) and 805R (5’-GACTACHVGGGGTATCTAATCC-3’) by a thermocycler PCR system. The program for PCR was 1 cycle of 98°C for 30s, followed by 32 cycles of denaturing at 98°C for 10s, annealing at 54°C for 30s, extension at 72°C for 45s and a final extension at 72°C for 10 min. Prior to 16S rDNA data analysis, the raw reads were demultiplexed, quality-filtered by fqtrim (V0.94) and merged by FLASH version 1.2.7.

Operational taxonomic units (OTUs) with 97 % sequence similarity using Vsearch (v2.3.4), and representative sequences for each OUT were determined through the Ribosomal Database Project classifier. Rarefaction analysis, alpha diversities (ACE, Chao, Shannon and Simpson indices), and the Weighted UniFrac nonmetric multidimensional scaling (NMDS) spearman coefficient were performed with the online platform at https://cloud.majorbio.com/.

Round 2

Reviewer 1 Report

The manuscript has been sufficiently improved.

Author Response

The manuscript has been sufficiently improved.

Reviewer 2 Report

Dear authors,

Thank you for submitting your rebuttal, and for addressing the points raised. Most points are handled in a satisfactory way. However, there are still a few points that need attention:

  • Figure and 5 and 6: Although a post-hoc test is done for each taxon, it seems there is no correction for the total number of genera tested. This is crucial in the type of explorative statistics provided here. In addition, the addition to the legend (*, ** and ***) is put with fig 5 rather than 6.
  • Figure 6: still no distance metric mentioned; it is in the rebuttal text but should be in the manuscript (either in methods or in the figure).
  • I would like to encourage the authors to submit the reads in the appropriate reads archive (for example ENA: https://www.ebi.ac.uk/ena/browser/submit).
  • “OUT” on line 178 should be “OTU".
  • I am not completely satisfied with the DAI explanation (figure 2c, line 230). How do the authors explain that DAI increases (or starts higher, based on the figure) from day 1 in the experimental groups, while the animals have not yet received DSS, and thus all treatments are the same between control and experimental groups? This implies that the compounds might have an effect. This is not explained. Could the authors elaborate on this?

Author Response

Dear authors,Thank you for submitting your rebuttal, and for addressing the points raised. Most points are handled in a satisfactory way. However, there are still a few points that need attention:

• Figure and 5 and 6: Although a post-hoc test is done for each taxon, it seems there is no correction for the total number of genera tested. This is crucial in the type of explorative statistics provided here. In addition, the addition to the legend (*, ** and ***) is put with fig 5 rather than 6.

Response: Thanks for your suggestions. It has been corrected.

• Figure 6: still no distance metric mentioned; it is in the rebuttal text but should be in the manuscript (either in methods or in the figure).

Response: Thanks for your suggestions. Distance-based Redundancy Analysis (db-RDA) plot showed the relationship between UC-related parameters and gut microbiota structure. The plot represents a db-RDA ordination using the Bray-Curtis distance. The black triangle represents the top ten species. We have been added to the figure caption.

• I would like to encourage the authors to submit the reads in the appropriate reads archive (for example ENA: https://www.ebi.ac.uk/ena/browser/submit).
Response: Thanks for your suggestion. The data presented in this study are available on request from the corresponding author.

• “OUT” on line 178 should be “OTU".Response: Thanks for your correction. It has been corrected.

• I am not completely satisfied with the DAI explanation (figure 2c, line 230). How do the authors explain that DAI increases (or starts higher, based on the figure) from day 1 in the experimental groups, while the animals have not yet received DSS, and thus all treatments are the same between control and experimental groups? This implies that the
compounds might have an effect. This is not explained. Could the authors elaborate on this?

Response: Thank you for your comments. I'm sorry for the misunderstanding. The figure shows the data on the first day after intragastric administration of 3% DSS. Since DSS is an induce for acute UC, symptoms are obvious and DAI is elevated a day later. Mice were given 3% (w/v) DSS for 7 days.
